# The Past, Present and Better Future of Feedback Learning in Large Language Models for Subjective Human Preferences and Values

**Hannah Rose Kirk**[1‡]**, Andrew M. Bean**[1]**, Bertie Vidgen**[1]**, Paul Röttger**[2]**, Scott A. Hale**[1,3]
[1]University of Oxford, [2]Bocconi University, [3]Meedan
[‡]hannah.kirk@oii.ox.ac.uk

## Abstract

Human feedback is increasingly used to steer the behaviours of Large Language Models (LLMs). However, it is unclear how to collect and incorporate feedback in a way that is efficient, effective and unbiased, especially for highly subjective human preferences and values. In this paper, we survey existing approaches for learning from human feedback, drawing on 95 papers primarily from the ACL and arXiv repositories. First, we summarise **the past**, pre-LLM trends for integrating human feedback into language models. Second, we give an overview of **present** techniques and practices, as well as the motivations for using feedback; conceptual frameworks for defining values and preferences; and how feedback is collected and from whom. Finally, we encourage a **better future** of feedback learning in LLMs by raising five unresolved conceptual and practical challenges.

## 1 Introduction

Incorporating human feedback into Large Language Models (LLMs) is a welcome development to create models that are better aligned with human preferences or values, and exhibit traits such as helpfulness, honesty and harmlessness (Askell et al., 2021; Bai et al., 2022a) or safety, quality and groundedness (Thoppilan et al., 2022). However, learning from human feedback introduces new biases and challenges, and there are many unresolved questions in this fast-moving field of research. It is important to take stock of current practices, possible blindspots, and new frontiers of research, so that tangible progress can continue to be made. In this paper, we adopt the dual aim to both survey existing literature on human feedback learning, then draw on the regularities, commonalities and

critiques of this survey to also provide recommendations for future work. We review 95 articles that use human feedback to steer, guide or tailor the behaviours of language models. This includes making models' responses more coherent and engaging (Lu et al., 2022); assisting models to infer user intent (Ouyang et al., 2022); rejecting and rebutting unsafe requests (Ganguli et al., 2022; Bai et al., 2022a); or minimising the risk of hallucination (Nakano et al., 2021; Glaese et al., 2022). We source articles primarily from the ACL and arXiv repositories, coding each according to a detailed conceptual and methodological schema. Our review makes three contributions:

- **The Past**: We include articles released both before and after the advent of LLMs, which avoids recency bias and allows us to track developments through time.
- **The Present**: We summarise current practices for incorporating human feedback learning into LLMs, such as reinforcement learning fine-tuning, supervised fine-tuning, and pre-training. We also document how feedback is collected and from whom.
- **The Future**: We draw on the findings of our review to highlight five unresolved challenges in the field; two challenges are conceptual and three are practical. Conceptual challenges revolve around the fundamental difficulty of specifying a clear shared set of preferences and values. And, even if the conceptual challenges can be resolved, practical challenges remain for converting abstract concepts into reliable signals to guide model behaviours.

We find that current processes of incorporating human feedback into LLMs often rely on unsatisfactory simplifying assumptions about the stabil-

ity, universality and objectivity of human preferences and values. What counts as a "good", "high-quality", "preferred" or "value-aligned" output is only objective in the abstract (Kirk et al., 2023a); so, we explicitly centre our review on subjective human preferences and values because we believe most text attributes retain some degree of contextual subjectivity. With this in mind, we call for more open, democratically-grounded and interdisciplinary collaboration, supported by robust processes of external scrutiny, to decide how different voices shape current and future LLMs.

## 2 Methods

### 2.1 Selecting Articles

We use a semi-automated method, casting a wide net of keywords to retrieve articles, then manually assessing their relevance for our review (see Tab. 1 for keywords and Appendix A for a schema).

**Initial set ($S_0$)** We retrieve articles from two corpora. First, we download the ACL anthology as a `.bib` file. Second, we use the arXiv API with the computation and language subclass (cs.CL) to find new or industry-led preprints that are not peer-reviewed but have early impact on the field. We match titles with $\geq 2$ keywords ($n = 187$), and deduplicate dual-posted articles ($n = 175$).[1]

**Inclusion criteria** Two authors read the abstract and introduction of $S_0$ articles, and included them if all the following questions were answered 'yes':
1. **Topic**: *Does the article seek alignment or adaptation of AI systems to human preferences and values?* This criterion excludes articles that functionally align aspects of language models e.g., word embedding alignment or sequence alignment.
2. **Modality**: *Does the article discuss language agents or language as its primary modality?* This criterion excludes any multimodal models, delegate agents or games-based RL.
3. **Empirical**: *Does the article contain empirical analysis or artefacts (such as experiments, datasets or models)?* This criterion excludes opinion papers, reviews or policy frameworks.

To ensure consistency, both authors coded the same 70 articles, finding 82% agreement in inclusion

decisions. We then discussed and refined the criteria before continuing. In total, 57 articles were included from $S_0$.

**Snowballed set ($S_1$)** To address blindspots in our keywords and corpora, we gather additional articles referenced within $S_0$ articles, regardless of where they were published ($n = 72$), and then reapply the inclusion criteria to ensure relevance. This results in 38 additional articles from $S_1$.

We further narrow our scope with two categorisations on the 95 articles from $S_0 + S_1$:

**Dominant contribution types** We categorise articles into: (i) those that *evaluate* or benchmark model's capabilities, ethics or worldviews ($n = 14$); (ii) those that *predict* human preferences and values from social media data using specialised models not intended for other downstream generative or classification tasks ($n = 9$); and (iii) those that *train* or seek to align models with a general notion of human preferred or valued text ($n = 72$). The last category is the focus of our review.[2]

**Use of LLMs** For the purposes of our review, we define LLMs as any encoder-only, decoder-only or encoder-decoder model that is pre-trained with self-supervised learning over large internet corpora. As a rule of thumb, we consider BERT (Devlin et al., 2019) and ELMO (Peters et al., 2018) among the first LLMs; so, any articles published before 2018 fall outside our definition. Of the 72 *train* articles, we cover 22 articles published in the pre-LLM era in our review of The Past (§3) and 50 articles using LLMs in The Present (§4).

### 2.2 Coding Articles

We examine each article under two main themes.[3] The **Conceptual** theme documents the motivations for collecting human feedback; the definitions of human preferences or values; whether these are understood as universal or contextual/cultural; and the level of interpretative freedom in applying preference or value concepts. The **Methodological** theme covers sub-categories on (i) annotation or labour force details, such as how feedback was collected and from whom; and (ii) modelling details, such as how feedback is integrated into training and evaluation phases, and the target task. We also

---

[1]We match on title to increase relevancy and because some ACL articles lack abstract metadata. In a sample of 100 retrieved articles, we determined that $\geq 2$ keywords best balanced relevancy with the size of the retrieved set. The cut-off for our automated collection is 17/02/2023.

[2]There is overlap between categories—papers that fine-tune LLMs according to human preferences also evaluate these trained models. See Appendix D for further detail.

[3]The full code book is presented in Appendix B.

| Keywords | Stemmed Keywords |
| --- | --- |
| alignment, human, value, moral, ethic, feedback, reinforcement, instruction, red teaming, red-teaming, preferences, harm, honest, helpful, personalis, personaliz | align, human, value, moral, ethic, feedback, reinforc, instruct, red team, red-team, prefer, harm, honest, helpful, personalis, personaliz |

Table 1: Keywords for retrieving articles from ACL and arXiv repositories. Highlighted keywords were not stemmed due to irrelevant matches e.g., "value" as "valu" returns many false positives including the title word "evaluation".

collect procedural details on academia vs industry authorship, whether empirical artefacts (data and/or models) are publicly available, and if (to the best of our knowledge) the paper has been peer-reviewed.

## 3 The Past

In this section, we review 22 articles released between 2014-2019 that use human feedback but with older generation model architectures. Highlighting these works ensures that foundational research is adequately attributed for advancements in today's models, and demonstrates the evolution from indirect or proxied human feedback.

### 3.1 Conceptual Classification

None of the articles released in this period seek alignment to human values. Instead, they generate text according to human preferences in machine translation (Mirkin et al., 2015; Mirkin and Meunier, 2015; Lawrence et al., 2017; Nguyen et al., 2017; Rabinovich et al., 2017; Kreutzer et al., 2018) and dialogue (Li et al., 2016; Mo et al., 2016; Li et al., 2017b; Wang et al., 2017; Liu et al., 2018; Jaques et al., 2019). Preferences are defined in both personal and universal contexts, reflecting the persistent difficulties of separating the two. Ficler and Goldberg (2017) focus on modulating formality depending on context, while others focus on the personalisation of language models, such as reflecting author personality in machine translation (Mirkin et al., 2015; Mirkin and Meunier, 2015; Rabinovich et al., 2017); providing financial recommendations via chat bots (Den Hengst et al., 2019); or enabling customised online shopping (Mo et al., 2016). Most studies target human preferences assumed to be commonly-held and stable, such as word order (Futrell and Levy, 2019), sense making (De Deyne et al., 2016; Seminck and Amsili, 2017) and vocabulary matching (Campano et al., 2014; Dubuisson Duplessis et al., 2017). In contrast, Nguyen et al. (2017) and Kreutzer et al. (2017) acknowledge the noisiness of human feedback but attempt to extract a single, unified preference.

### 3.2 Methodological Classification

Most articles use pre-transformer recurrent neural networks such as LSTMs (Hochreiter and Schmidhuber, 1997; Vaswani et al., 2017). Few articles use direct human feedback, mostly in information retrieval tasks. In two cases, humans answer a series of yes/no questions to provide a more expressive reward for reinforcement learning (RL) (Li et al., 2017a; Lawrence and Riezler, 2018). Dhingra et al. (2017) use requests for additional information to form better queries with a binary 'success/failure' reward. Lawrence et al. (2017) and Lawrence and Riezler (2018) compare forms of human feedback, finding cardinal feedback to be more useful than pairwise comparison.

Human feedback is an expensive and time-consuming source of data to collect, which motivates efforts to find reliable proxies (Lawrence et al., 2017; Nguyen et al., 2017). Implicit feedback methods attempt to utilise naturally-occurring signals in human interactions, such as sentiment (Wang et al., 2017; Jaques et al., 2019) and response length (Campano et al., 2014). Other articles define rules on desirable dialogue properties, such as length (Li et al., 2016), vocabulary alignment (Dubuisson Duplessis et al., 2017), or tone (Ficler and Goldberg, 2017), and score the agent for achieving them. Only Li et al. (2016) apply RL to further train the model from the feedback.

Simulating human feedback is also a commonly-used and cost effective approach where 'feedback' is generated by measuring similarity to the gold standard in pre-existing, human-labelled datasets. Parallel translation corpora are a common source of gold demonstrations, e.g., translated TED talks (Mirkin et al., 2015; Mirkin and Meunier, 2015; Nguyen et al., 2017) or European Parliament speeches (Kreutzer et al., 2017; Rabinovich et al., 2017). User simulators typically use a 'success/failure' score for RL (Mo et al., 2016; Liu et al., 2018), while 'gold standard' approaches use a more complex loss function on output similarity (Mirkin and Meunier, 2015; Nguyen et al., 2017).

## 4 The Present

Turning our attention to the heart of our review, this section discusses the 50 papers that incorporate human feedback to steer LLM behaviours.

### 4.1 Conceptual Classification

We first seek to understand **why human feedback is collected**. The motivations for eliciting feedback form two groups. The first group generally seeks *value alignment*, i.e., some notion of steering language models towards producing societally-desirable text (Zhao et al., 2021; Liu et al., 2021a). We note a variety of vague goals such as to reduce "non-normative" (Peng et al., 2020) or "immoral" text (Liu et al., 2023c); to generate more "pro-social" (Liu et al., 2022) or "legitimate" text (Bakker et al., 2022); or to encourage that LLM technologies have a "positive impact on society" (Liu et al., 2023b). Specific motivations include minimising toxic or offensive language (Dinan et al., 2019; Xu et al., 2021a; Ju et al., 2022; Scheurer et al., 2022; Korbak et al., 2023); improving safety (Liu et al., 2021a; Xu et al., 2021b; Thoppilan et al., 2022; Ganguli et al., 2022; Jin et al., 2022); adapting to ethical or moral scenarios (Forbes et al., 2020; Jiang et al., 2022; Jin et al., 2022); or achieving political ideological balance (Liu et al., 2021b). The broad definitions of value alignment mostly assume some universality of value dimensions.[4] However, some do seek to align LLMs to specific groups, sets of values or according to cultural context (Solaiman and Dennison, 2021; Qiu et al., 2021; Bang et al., 2022).

The second group of articles is motivated by more practical target concepts of improving model capabilities, particularly when clear optimisation metrics or programmatic rewards are lacking (Ziegler et al., 2019; Wu et al., 2021; Glaese et al., 2022; Bai et al., 2022b). Motivations often revolve around generating high-quality or human-preferred outputs (Gao et al., 2018; Böhm et al., 2019; Jaques et al., 2020; Stiennon et al., 2020; Wang et al., 2021; Scheurer et al., 2022; Nguyen et al., 2022; Xu et al., 2022), without much discussion of why this matters or whether humans agree amongst themselves what is "high-quality". Specific target attributes include minimising repetitiveness (Arora

et al., 2022); increasing coherence (Lu et al., 2022), usefulness (Liu et al., 2021a), engagingness (Gao et al., 2020; Xu et al., 2021b; Lu et al., 2022), or interest (Thoppilan et al., 2022); and producing human-like conversations (Hancock et al., 2019; Jaques et al., 2020). Some seek greater explainability and factuality in generated text (Nakano et al., 2021; Menick et al., 2022; Scheurer et al., 2022; Thoppilan et al., 2022) or correctness in code (Korbak et al., 2023). Preferences can also be elicited for customisation and personalisation (Majumder et al., 2019; Zhou et al., 2021; Deng et al., 2022).

The boundary between preference- and value-motivated aims is not always clear-cut. Commonly-adopted mixed motivations include helpful, honest and harmless behaviours, introduced by Askell et al. (2021) and adopted by others (Bai et al., 2022b,a; Bakker et al., 2022; Menick et al., 2022). Thoppilan et al. (2022) target safety, quality and groundedness—concepts that similarly blur the lines between practical preferences and value-laden judgements. Even for instruction-tuning articles motivated by inferring user intent, what Ouyang et al. (2022) call "explicit" and "implicit" intent is synonymous with the helpfulness versus honesty/harmlessness distinction.[5]

### 4.2 Methodological Classification

We primarily discuss how feedback is collected (§4.2.1) and integrated into LLMs (§4.2.2). We additionally present an overview of target tasks and evaluation methods in Appendix C.

### 4.2.1 Collecting Feedback

First, we address **how feedback is collected**. Explicit comparisons collected on model outputs are used to reveal the preferences of human raters (Gao et al., 2018; Ziegler et al., 2019; Askell et al., 2021; Jaques et al., 2020; Stiennon et al., 2020; Ganguli et al., 2022; Glaese et al., 2022).[6] More fine-grained feedback includes binary or Likert scale questions on text attributes (Nakano et al., 2021;

---

[4]An example of a vague definition is that a "value-aligned system should make decisions that align with human decisions in similar situations and, in theory, make decisions which are unlikely to be harmful" (Nahian et al., 2020, p. 1).

[5]Ouyang et al. (2022, p.2) include "explicit intentions such as following instructions and implicit intentions such as staying truthful, and not being biased, toxic, or otherwise harmful."

[6]Usually, ratings are collected between two outputs (Bai et al., 2022b; Ganguli et al., 2022) but others use four (Ziegler et al., 2019) or even up to 9 items for comparison (Ouyang et al., 2022). A null vote is predominately not included (*neither of these outputs are good*) which may be a particular problem for harm assessment (Ganguli et al., 2022)—though some address ties in preference strength (e.g., Bai et al., 2022a; Menick et al., 2022).

Menick et al., 2022; Thoppilan et al., 2022); natural language comments (Ju et al., 2022; Scheurer et al., 2022); or edits (Hancock et al., 2019; Lu et al., 2022; Liu et al., 2023c). Ideal demonstrations are used to ground norm-dependent or ethical judgements (Forbes et al., 2020; Zhao et al., 2021; Pyatkin et al., 2022; Jiang et al., 2022; Jin et al., 2022), or in combination with ratings to prime model behaviour (Nakano et al., 2021; Wu et al., 2021; Ouyang et al., 2022; Bakker et al., 2022). Several articles collect negative feedback via adversarial examples (Dinan et al., 2019; Xu et al., 2021a,b; Glaese et al., 2022). Xu et al. (2022) test various feedback types including binary, free-form conversation, and fine-grained failure modes.

Human input can be further removed from directly assessing model outputs. For example, simulating feedback with an "oracle" assumed to prefer specific text attributes measured via automated metrics (Wang et al., 2021; Nguyen et al., 2022; Korbak et al., 2023) or predictions from a separate classifier (Peng et al., 2020; Liu et al., 2021b). In Bai et al. (2022b) human input defines the constitution but AI feedback is used to implement it during training. A seed of human generated examples guiding synthetic data generation also applies elsewhere (Bang et al., 2022; Castricato et al., 2022; Honovich et al., 2022; Wang et al., 2022). Other articles adopt human labels on pre-existing datasets (Böhm et al., 2019; Liu et al., 2021a; Arora et al., 2022; Jiang et al., 2022), or leverage implicit feedback data from stories (Nahian et al., 2020) and social media such as Reddit or StackOverflow (Gao et al., 2020; Askell et al., 2021; Bai et al., 2022a). Feedback can also be inferred from certain language patterns or emotive attributes in conversations with human partners (Hancock et al., 2019; Zhou et al., 2021).

Second, we address **who feedback is collected from**. Almost all articles use crowdworkers for training and/or evaluation, recruited from a variety of sources—including MTurk (Nahian et al., 2020; Peng et al., 2020; Jaques et al., 2020; Liu et al., 2021a,b; Qiu et al., 2021; Bai et al., 2022a; Ganguli et al., 2022; Jin et al., 2022; Xu et al., 2022; Ju et al., 2022); Upwork (Stiennon et al., 2020; Bai et al., 2022a; Ganguli et al., 2022); ScaleAI (Ouyang et al., 2022; Stiennon et al., 2020; Ziegler et al., 2019); and SurgeAI (Solaiman and Dennison, 2021; Nakano et al., 2021; Bai et al., 2022b). With 'in-the-wild' social media data, social media users unknowingly become the 'raters' (Gao

et al., 2020; Askell et al., 2021; Bai et al., 2022a). Ouyang et al. (2022) include OpenAI API users as "demonstrators". At least 13 articles rely on their authors for a variety of tasks:[7] writing seeds to scale synthetic data (Honovich et al., 2022; Wang et al., 2022); hand-crafting conditioning prompts (Askell et al., 2021; Glaese et al., 2022); defining a constitution (Bai et al., 2022b); specifying topics or starter questions (Solaiman and Dennison, 2021; Bakker et al., 2022), and ethical scenarios (Zhao et al., 2021); conducting evaluation (Stiennon et al., 2020; Ganguli et al., 2022; Lu et al., 2022) or generating benchmarks (Bai et al., 2022a); and compiling training tasks for crowdworkers (Qiu et al., 2021; Glaese et al., 2022). Even without direct involvement, authors can influence feedback collection by writing annotator guidelines.

### 4.2.2 Integrating Feedback

**RL with Direct Human Feedback** A reward signal can first be extracted by asking *actual humans* about their preferences for model outputs then embedded into the LLM via RL fine-tuning. The general recipe is as follows: **(Step 1)**: Either take an "off-the-shelf" pre-trained LLM (Lu et al., 2022); Or adapt this model via prompt-guiding (Askell et al., 2021; Bakker et al., 2022; Bai et al., 2022a; Glaese et al., 2022) or supervised fine-tuning and behavioural cloning over ideal demonstrations (Ziegler et al., 2019; Stiennon et al., 2020; Nakano et al., 2021; Ouyang et al., 2022; Menick et al., 2022).[8] **(Step 2)**: Generate multiple outputs from this model, and employ crowdworkers to create a comparisons dataset. **(Step 3)**: Train a preference reward model (PM) on this feedback so "better" items are assigned higher score (Bai et al., 2022a)—either a scalar reward for a given item or an ELO score i.e., the log odds that A ≻ B (Stiennon et al., 2020; Nakano et al., 2021; Glaese et al., 2022). The PM can be pre-trained

---

[7]In Scheurer et al. (2022), the article relies on two authors to provide the feedback data and two other authors to do the human evaluation experiments.

[8]For example, Lu et al. (2022) use a SOTA chinese chatbot; Nakano et al. (2021) start with GPT-3 architecture (760M, 13B, 175B); Bai et al. (2022a) use the Context-Distilled LM (52B) from Askell et al. (2021); Glaese et al. (2022) handauthor prompts to demonstrate 'good' behaviour in a Dialogue Prompted Chincilla model (70B); Stiennon et al. (2020) start with versions of GPT-3 (1.3B, 6.7B) fine-tuned on filtered TL;DR Reddit dataset; Ziegler et al. (2019) use a fine-tuned GPT-2 model; Menick et al. (2022) use supervised fine-tuning on Gopher family models with examples rated positively by labellers; and Ouyang et al. (2022) fine-tune GPT-3 on demonstrations of desired behaviours.

on naturally-occurring text and ratings e.g., from Reddit or Stackoverflow (Askell et al., 2021; Bai et al., 2022a). **(Step 4)**: Fine-tune a RL policy (another LM) that generates text autoregressively, whilst the PM provides a reward signal. Often, the policy is updated using the PPO algorithm (Ziegler et al., 2019; Stiennon et al., 2020; Nakano et al., 2021) and a KL-penalty term is applied to control deviations from the base model (Jaques et al., 2019; Ziegler et al., 2019; Stiennon et al., 2020; Nakano et al., 2021; Menick et al., 2022; Ouyang et al., 2022; Liu et al., 2023c). This pipeline can be implemented in offline, online or batched settings (see Ziegler et al., 2019). Modifications to the recipe include using recursive subtasks (Wu et al., 2021); applying a rule reward model in addition to the PM to penalise undesired outputs (Glaese et al., 2022); or using the PM to re-rank or reject sample generations from the supervised model (Askell et al., 2021; Nakano et al., 2021; Glaese et al., 2022; Ganguli et al., 2022; Bai et al., 2022a; Xu et al., 2022; Bakker et al., 2022), which can match or outperform optimising a model via RL (Menick et al., 2022; Thoppilan et al., 2022).

**RL with Indirect Human Feedback**   A reward can be *inferred* without directly asking humans about their preferences over model outputs. These articles skip the step of training a PM from comparisons data, and instead infer preferences from textual attributes of outputs (Jaques et al., 2020; Zhou et al., 2021). It varies how far removed the human input is, for example in designing the constitution (Bai et al., 2022a), in determining the automated metric (Nguyen et al., 2022; Korbak et al., 2023) or in compiling the word lists to measure political bias (Liu et al., 2021b). Often another model is treated as the 'oracle' to simulate human rewards—Gao et al. (2018), for example, simulate preferences on two summaries with perfect, noisy and logistic noisy "oracles" based on ROGUE scores; Wang et al. (2021) take the reward as human revisions from parallel machine translation corpora; while others deploy the rewards from a value, moral or toxicity classifier trained on crowdworker labels to reinforce a generator (Qiu et al., 2021; Liu et al., 2022; Castricato et al., 2022; Pyatkin et al., 2022).

**Generator and Discriminator**   Some use a unified generator and classifier step to steer the LLM away from undesirable text (Arora et al., 2022), for example using other fine-tuned LLMs to modify the predicted probability in a base model for the next token at decoding time (Liu et al., 2021a). A combined model that functions as a generator and a discriminator can be trained sequentially (Thoppilan et al., 2022) or jointly (Lu et al., 2022).

**Preference Pre-training**   Korbak et al. (2023) argue that incorporating human feedback in supervised or RL fine-tuning phases is suboptimal. Instead, they approach alignment in the pre-training phase of GPT-2, finding that conditional training is the most effective pre-training objective, and is more robust than later fine-tuning an already pre-trained model.

**Preference Fine-Tuning**   Human feedback can be incorporated via supervised fine-tuning (Hancock et al., 2019; Nahian et al., 2020; Jiang et al., 2022). For example, Gao et al. (2020) apply contrastive learning with a GPT-2 based dialogue model over 133M pairs of human feedback data with a loss designed to simultaneously maximise the positive sample score and minimise the negative score. Liu et al. (2023b) use "chain of hindsight" fine-tuning to include both positive and negative feedback. Fine-tuning data is often filtered relative to the value or preference goal (Solaiman and Dennison, 2021; Xu et al., 2022; Bang et al., 2022). Peng et al. (2020) instead train a reward model (normative text classifier) but this reward is applied to the loss and backpropagated during fine-tuning.

**Prompting**   Prompting is a simple way to align LLMs with specified human preferences and values. Jin et al. (2022) cast moral situations as multi-step prompts to elicit chain of thought reasoning in InstructGPT, while Zhao et al. (2021) use zero- and few-shot prompts for responsive questioning on unethical behaviours. Askell et al. (2021) show that using a long prompt (4,600 words) from ideal author-written conversations is an effective alternative to fine-tuning in data-constrained scenarios. They also use context distillation by training a new LLM to replicate the behaviour of another LLM that is using a specific prompt.

## 5   Challenges and Recommendations for the Future

Drawing on our analysis of the reviewed papers, we identify five key challenges for future researchers. These challenges are divided into conceptual and practical issues. The conceptual challenges (C1-C3) revolve around the difficulty of specifying a

clear set (or sets) of preferences and values. Even assuming the resolution of the conceptual challenges, practical challenges remain in converting conceptual ideals into empirical signals, which in turn steer language model behaviours.

**(C1) Preferences and values are not universal** 'Aligning' a language model requires a set of desired preferences or values to align with; but specifying such a set is an unresolved problem. One popular approach is to specify a minimal set of ostensibly unobjectionable and widely-shared values, such as helpfulness, honesty and harmlessness (Bai et al., 2022a,b; Thoppilan et al., 2022). However, these values are only unobjectionable because they are abstract and not precisely defined (Kirk et al., 2023a). These terms can be considered what Levi-Strauss and Laclau call 'empty signifiers' (Lévi-Strauss, 1987; Laclau, 2005); terms that are viewed positively but are inscribed with different meanings by different people. For example, when Bai et al. (2022a) design a constitution to produce outputs as "ethical and harmless as possible", this can have varying interpretations based on an individual's own ethical frameworks and sociocultural background. Establishing priorities over sets of preferences or values to embed in LLMs, and ensuring consistent interpretation of conceptual meaning across people, is a persistent challenge which cannot alone be resolved via purely technical solutions. One possible approach is to draw on legal theory, and values protected in human rights law (Solaiman and Dennison, 2021). Translating abstract shared values into decisions is a core function of legal systems and legal theory offers a long history of scholarship which combines the philosophical and practical. One approach along these lines was proposed by Kirk et al. (2023b) which applies a principle of subsidiarity to govern the personalisation of generative AI systems for different use cases. We also advocate for anchoring closely to existing legal systems as a matter of democratic principle: it is dangerous for moral and value judgements with broad societal impacts to be made by small independent groups.

**(C2) Preferences and values are inconsistently defined** Although the terminology of 'preferences' and 'values' implies some difference between the two, the conceptual basis and normative implications of this distinction is often unclear. Colloquially, values are understood to be stronger than preferences, and potentially carry greater normative weight as guiding principles or life goals (Fischer, 2017). As such, users may have greater concerns about an LLM misaligned with their values than with their preferences; So, it is important to be clear about which is being discussed. Within the broad terms, there are many meanings: 'preferences' have been defined as 'instrumental utility' (Dubuisson Duplessis et al., 2017; Gao et al., 2018; Nguyen et al., 2022), 'stylistic taste' (Mirkin and Meunier, 2015; Seminck and Amsili, 2017; Jaques et al., 2020), and 'behavioural principles' (Bai et al., 2022b; Castricato et al., 2022). 'Values' definitions are based on 'instrumental and intrinsic value' (Askell et al., 2021), 'acceptable social behaviours' (Forbes et al., 2020; Bang et al., 2022), or 'making decisions which are unlikely to be harmful' (Nahian et al., 2020). The differences between individual (subjective) and global (objective) preferences is often blurred—for example, which properties of a "better" summary are universal, and which depend on subjective appreciation, like writing style and tone. Clearer definitions of preferences and values in the context of alignment would serve to motivate and clarify *what* we are aligning LLMs to.

**(C3) Human feedback is inherently incomplete** Alignment via human feedback ultimately relies on LLMs being capable of successfully generalising from few examples to new cases and domains. This is because the space of possible behaviours over which to collect feedback is prohibitively large and is not fully known. An open question is the extent to which models generalise from partial human feedback, especially when presented with data that is completely out-of-domain for their training or at the margins of its distribution. For instance, if an LLM is trained with examples of safe responses to user prompts which deny the Holocaust, it may generalise to different expressions of the same canonical request. However, it will not necessarily learn how to handle denials that the earth is round and denials of vaccine efficacy, or have domain expertise for other harmful requests, such as users who ask how to make a bomb or bio-weapon. Human values are considered to be fairly stable guiding principles that manifest similarly across situations for a given individual (Fischer, 2017) but the same generalisation cannot be guaranteed of LLMs.

Several related epistemological issues arise from technical details of the methods being used. Rein-

forcement learning introduces a path-dependence problem, where the particular order in which feedback is given may change the quality of the final results. As a result, it is difficult to know whether a local optimum is reached which is notably worse than the global optimum. With any form of learning from feedback, language models may also overfit or appear to be aligned externally, but have persistent internal misalignment which manifests subtly in cases more distant from the training data (Perez et al., 2022). These challenges become yet more convoluted when dealing with more complex tasks—an issue that Bowman et al. (2022) examine in their discussion of scalable oversight.

**(C4) Operationalising a "good" output is difficult**   Even if a shared set of values could be agreed upon, converting these thick normative concepts into signals that models can use, such as by collecting annotator ratings, is hard. Complex goal operationalisation is itself a motivator for collecting feedback—when humans may not be able to articulate their preferences or write ideal demonstrations but can rate outputs, a kind of "I know it when I see it" logic. However, training with human feedback involves moving values and preferences from the abstract to particular survey or rating instruments, reinforcing differences in interpretation. To reduce disagreements, some authors write very prescriptive and/or comprehensive guidelines for the task in order to "make comparisons as unambiguous as possible" (Nakano et al., 2021, p.18). Several papers still find low inter-annotator agreement with such prescriptive approaches (Stiennon et al., 2020; Glaese et al., 2022; Bai et al., 2022a; Ouyang et al., 2022). In other cases, annotators are explicitly allowed to use their own subjective assessment, to "interpret these concepts as they see fit" (Bai et al., 2022a, p.4), but then agreement between annotators is no longer ensured. When multiple text attributes affect annotators' preferences, it is hard to pin down what we are actually measuring. For example, Stiennon et al. (2020) and Wu et al. (2021) condition their evaluation question as "how good is this summary, given that it is X words long?". Hypothetically, if "good" is subjective then the question should be "how good is this summary for individual Y?". Some guidelines do ask annotators to role-play or put themselves in the shoes of others, for example to infer the intent of a prompt (Ouyang et al., 2022) or question (Nakano et al., 2021), but this may introduce further problems, especially for value-laden judgements where the rater may have a biased interpretation of how to apply another person's values (Qiu et al., 2021).

To aid transparent communication, it should be clearly documented whether researchers aspire to follow the prescriptive or subjective paradigm of data annotation, rather than leaving it unspecified (Röttger et al., 2022; Kirk et al., 2023a). Increased interdisciplinary communication with practitioners in other fields would impart wisdom on measuring the perspectives and behaviours of human subjects. For example, Human-Computer Interaction literature shows how interfaces and incentives can be optimally designed to avoid participant response bias (Deng and Poole, 2010; Dell et al., 2012; Hsieh and Kocielnik, 2016); Experimental psychology and behavioural economics research show how the presentation of scales and order effects influence ratings (Friedman et al., 1994; Maeda, 2015; Westland, 2022) and that human preferences are unstable, intransitive and vulnerable to experimental artefacts (Tversky, 1969; Lacy, 2001; Chiesa and Hobbs, 2008; Lee et al., 2009; Chuang and Schechter, 2015). Researchers should consider techniques to model the noise and distribution of human feedback (Ju et al., 2022) or establish post-hoc consensus (Bakker et al., 2022), rather than ignoring disagreement by aggregating responses. However, there are trade-offs: the specific nuances and minutiae captured in fine-grained feedback might heighten biases and reduce generalisability when drawn from unrepresentative samples—which we now discuss.

**(C5) Crowdworkers and social media users are neither representative nor sufficient**   A degree of subjectivity persists even with prescriptive guidelines and well-designed experimental instruments; So, outcomes critically depend on who is interpreting value or preference-based concepts. In the majority of articles, fewer than 100 humans are employed to guide or evaluate language model behaviours (Jaques et al., 2020; Stiennon et al., 2020; Nakano et al., 2021; Menick et al., 2022; Bai et al., 2022a; Ouyang et al., 2022; Jin et al., 2022; Pyatkin et al., 2022), which is concerning for ethically or morally ambiguous scenarios. It is striking that so few voices have so much power in shaping LLM behaviours—in Bai et al. (2022a) just 20 humans contributed 80% of the feedback data, and in Nakano et al. (2021) the top 5 humans contributed 50%. Workforces employed for evaluation are similarly small, with some em-

ploying <25 workers (Scheurer et al., 2022; Castricato et al., 2022; Gao et al., 2018; Liu et al., 2023b). Overwhelmingly, these humans are US-based, English-speaking crowdworkers with Master's degrees and between the ages of 25-34. This results in a non-democratic and non-diverse feedback process, termed "the tyranny of crowdworker" by Kirk et al. (2023b), which has been shown to introduce political and religious biases in model behaviours (Perez et al., 2022). The limitations of relying on the subjective interpretations of a small and non-representative work force are exacerbated by inadequate documentation. Only nine out of 50 papers provided solid documentation, such as demographic breakdowns (Stiennon et al., 2020; Thoppilan et al., 2022; Bai et al., 2022a; Ganguli et al., 2022; Glaese et al., 2022; Jin et al., 2022; Liu et al., 2022; Ouyang et al., 2022; Liu et al., 2023c). Others provide high-level details of the rater pool such as number of workers, hiring platform, or aggregate demographics. The majority of articles do not document their workforce, nor discuss sample biases or annotator artefacts.

When soliciting human feedback, attempts should be made to diversify who is given a voice, such as by applying democratic or jury-based principles in how these voices are weighted (Gordon et al., 2022) and by employing bottom-up participatory approaches (Martin Jr. et al., 2020; Birhane et al., 2022; Zytko et al., 2022; Derczynski et al., 2023); Or to seek top-down sampling that better represents the population being studied (Bakker et al., 2022). Mirroring the move in other areas of NLP to document and explore annotator disagreement (Aroyo and Welty, 2015; Geva et al., 2019; Nie et al., 2020; Prabhakaran et al., 2021; Davani et al., 2022), each item of feedback should be associated with a pseudo-anonymised annotator ID. So far as privacy allows, documentation of annotator background should be provided in a data statement (Bender and Friedman, 2018).

## 6 Conclusion

This review provided an overview of incorporating human feedback into LLMs, with a focus on subjective preferences and values that lack 'ground truth alignment'. We have witnessed two notable shifts in the field from past to present—first, a move away from specialist systems towards general purpose language agents capable of handling many NLP subtasks via instruction or open-ended dialogue;

second, more use of direct human feedback which surpasses the limitations of user simulations or automated metrics.

While the shift to incorporate human voices directly into LLM development is welcome, it introduces new challenges that require careful navigation. Some challenges are more tractable than others—for example, practitioners will always have to deal with the complexities and intricacies of unstable and idiosyncratic preferences across end users of their model, but can take practical steps to better approximate this distribution by diversifying the recruitment of feedback providers.

External scrutiny is crucial to ensure the integrity and reliability of research efforts. Our review shows that many influential papers lack open and externally-validated peer review, especially those published by big industry labs like Google DeepMind, Anthropic, Google, and OpenAI. Furthermore, the majority of reviewed papers do not release model artefacts, or only do so behind a paywalled API. To foster progress, we advocate for a greater degree of open, interdisciplinary and democratically-grounded discussion on how humans can meaningfully shape future LLM behaviours in a way which is well-bounded, operationalisable, and equitable.

## 7 Limitations

We discuss limitations associated with our review:

**Applying topic exclusion** We exclude articles on the basis of being unrelated to the topic of value or preference alignment, but found it consistently difficult to draw a clear distinction between articles in and out of scope. For validation purposes, we had both reviewers read a portion of the articles, and found the cases of disagreement helpful to highlight this challenge. One such example was with two articles using similar methods to approach translation which we initially classified differently, Kreutzer et al. (2017) and Lawrence et al. (2017). The papers primarily treat translation as an objective task focused on BLEU scores, which would make them out of scope. However, translation inherently involves stylistic and subjective judgements, and the methods developed seek to replicate these judgements from the training data, blurring the distinction. Honesty is another target concept with these issues—whether in-text claims are referenced is fairly black and white, but whether an end user ascribes more trust to the system be-

cause of these references is subject to idiosyncratic epistemology. We use this case to highlight the weaknesses of creating a dichotomy between subjective and objective preferences in practice.

**Related NLP subfields** A related issue is where to draw the line for what is and is not in scope. Some narrowing was needed to make the review focused, feasible and instrumentally useful to future practitioners. However, technically fairness and bias are values of intrinsic utility—hence their inclusion in many AI principles around the world (Jobin et al., 2019). That said, there exists a very wide and distinct literature on fairness and bias in LLMs that would be too expansive for this review (see, e.g., Chang et al., 2019; Lucy and Bamman, 2021; Abid et al., 2021; Nadeem et al., 2021; Kirk et al., 2021; Smith et al., 2022). Similarly, there are sub-literatures on other aspects of LLM behaviours—such as toxicity (Gehman et al., 2020; Welbl et al., 2021), truthfulness (Lin et al., 2022) or hallucination (Ji et al., 2022). We explicitly focus on papers that target some notion of human preferences and values in their motivations, but the challenges raised from our review can be applied to other fields which similarly suffer from subjectivity in interpretative scope—e.g., carefully deciding who the human labellers are and what guidelines govern their interpretation of concepts.

**Blindspots in reviewed articles** Blindspots come from a number of sources. First, *keyword and corpora blindspots*: We ground our initial review on articles from arXiv and ACL using a set of defined keywords. We attempt to mitigate blindspots by snowballing related and relevant articles outside our initial collection; however, it is almost certain that we have missed some papers in the field as a whole. Second, *language blindspots*: Our review only contains articles written in English, limited by the expertise of authors who acted as the coders. This bias however reflects the dominance of English in academic publishing in general, but English language proficiency may gatekeep the concepts and voices already contributing to LLM development. Third, *community blindspots*: we only look at academic papers—but issues surrounding large language model behaviours or alignment have become a hot topic of conversation on blogs and social media forums. We inherently exclude such discussions from these other stakeholder communities. Fourth, *modality blindspots*: there is a

rich history of using RL to align models in other modalities, such as delegate agents acting in toy or game worlds (see, e.g., Christiano et al., 2017). We do not cover the insights from these related literatures. Finally, *temporal blindspots*: research into LLMs is a fast-paced field—in one week, there can be as many as 500 articles posted on the cs.CL subclass of arXiv. Inevitably, other influential articles have been released after the review was completed and more were released during its peer review period. A good example of this is Rafailov et al. (2023) who introduce Direct Preference Optimisation, a technique that could substantially change how people approach feedback learning in the future. Other relevant papers that appeared after the cut-off for this review include Dong et al. (2023); Hosking et al. (2023); Liu et al. (2023d,a); Song et al. (2023); Yuan et al. (2023); Wu et al. (2023); Zhou et al. (2023). With the field's rapid developments, any review paper runs the risk of lagging behind the latest research. However, given the substantial number of articles that we did review, we expect many of the general findings and highlighted challenges to apply in upcoming future work.

**External scrutiny of reviewed articles** We consciously made the decision to include articles which have not yet been peer reviewed to stay ahead of the curve with early-released pre-prints and also to track industry contributions (which are often not externally peer reviewed). In the 22 papers appearing the The Past section, 18 were peer reviewed. Of the 50 papers appearing in The Present section, only 21 were clearly peer-reviewed. It is a contentious issue that many influential papers lack standard practices of external scrutiny and rigorous academic backstops, though often industry-authored papers do undergo a process of internal review before a preprint is released.

## Acknowledgements

This paper received funding from a MetaAI Dynabench grant as part of a research agenda on optimising feedback between human-and-model-in-the-loop. H.R.K.'s PhD is supported by the Economic and Social Research Council grant ES/P000649/1. A.M.B.'s PhD is supported by the Clarendon Fund. P.R. received funding through the INDOMITA project (CUP number J43C22000990001) and the European Research Council (ERC) under the European Union's Horizon 2020 research and innovation program (No. 949944, INTEGRATOR).

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

## A Flowchart of Articles for Scoping the Review

In Fig. 1, we schematically summarise the process of selecting articles for our review.

## B Code Book

We present the full code book used for each article in Tab. 3. These questions were inputted into an online form then coded by two authors of the paper, with frequent check-ins to ensure similarity of interpretation on how the form should be used. The first theme (conceptual) makes up our conceptual comments in the main paper, while the laboural and technical themes make up our methodological comments in the main paper.

## C Additional Information on Reviewed Articles

### C.1 Target Tasks

In Tab. 2, we summarise the core target tasks approached by each article. Reflecting the recent movement away from specialist NLP systems towards general purpose language agents, the majority of articles work with generalised models that can handle many other NLP subtasks via instruction or dialogue.

### C.2 Evaluating Models

Even articles employing indirect or simulated human feedback usually conduct a human evaluation stage (Peng et al., 2020; Liu et al., 2021b, 2022). Differently-trained models are often compared via ELO scores or win rates (Ziegler et al., 2019; Nakano et al., 2021; Bai et al., 2022b,a; Scheurer et al., 2022; Bakker et al., 2022; Glaese et al., 2022; Ouyang et al., 2022). Most evaluations include fine-grained questions about model outputs, including quality or usefulness (Wu et al., 2021; Nakano et al., 2021; Liu et al., 2022; Bakker et al., 2022); political bias (Liu et al., 2021b); coherence (Wu et al., 2021; Nakano et al., 2021; Liu et al., 2022; Bakker et al., 2022); safety or harmlessness (Xu et al., 2021a; Lu et al., 2022; Ganguli et al., 2022; Thoppilan et al., 2022); informativeness, correctness or trustworthiness (Wu et al., 2021; Nakano et al., 2021; Lu et al., 2022); creativity (Honovich et al., 2022); and alignment with a human value or trait (Solaiman and Dennison, 2021; Liu et al., 2022; Castricato et al., 2022; Liu et al., 2023b).

Others use automated metrics to quantitatively compare models and outputs, with Böhm et al. (2019) and Stiennon et al. (2020) performing a comparison of how such automated metrics correlate with human preferences. Metrics include ROGUE (Böhm et al., 2019; Ziegler et al., 2019; Stiennon et al., 2020; Liu et al., 2022; Nguyen et al., 2022; Wang et al., 2022; Wu et al., 2021; Liu et al., 2023c), summary length (Stiennon et al., 2020), perplexity (Liu et al., 2021b, 2022, 2023c) or Sacre-BLEU (Wang et al., 2021). Sometimes separate discriminative classifier are deployed to measure textual attributes (Thoppilan et al., 2022), such as toxicity measured via Perspective API scores (Solaiman and Dennison, 2021; Arora et al., 2022). Scheurer et al. (2022) score how close feedback and refinements are in the embedding space because they find written feedback often describes an "ideal" output. Any prediction tasks – e.g., whether an ethical judgement is fair or unfair (Jiang et al., 2022), a situation is normative or non-normative (Nahian et al., 2020; Forbes et al., 2020), a norm exception is permissible or not permissible (Jin et al., 2022) or an utterance is value aligned or misaligned (Qiu et al., 2021) – use F1-score or accuracy as evaluation metrics.

Metrics or human evaluations that measure how aligned a resultant model is with human prefer-

| Task | References |
|------|-----------|
| Text generation | (Peng et al., 2020; Liu et al., 2021b; Solaiman and Dennison, 2021; Arora et al., 2022; Liu et al., 2022; Korbak et al., 2023), *including story generation* (Castricato et al., 2022) and *code generation* (Korbak et al., 2023) |
| Instruction following | (Honovich et al., 2022; Ouyang et al., 2022; Wang et al., 2022) |
| Open-ended dialogue | (Hancock et al., 2019; Gao et al., 2020; Jaques et al., 2020; Askell et al., 2021; Qiu et al., 2021; Thoppilan et al., 2022; Bai et al., 2022b,a; Ganguli et al., 2022; Lu et al., 2022; Xu et al., 2022; Liu et al., 2023b), *including information-seeking dialogue* (Glaese et al., 2022) |
| Open-book generative QA | (Zhao et al., 2021; Deng et al., 2022; Nakano et al., 2021; Menick et al., 2022) |
| Summarization | (Gao et al., 2018; Böhm et al., 2019; Ziegler et al., 2019; Stiennon et al., 2020; Scheurer et al., 2022; Nguyen et al., 2022; Liu et al., 2023b), *including long-form book summarisation* (Wu et al., 2021) and *opinion consensus summarisation* (Bakker et al., 2022) |
| Toxic language | (Dinan et al., 2019; Peng et al., 2020; Liu et al., 2021a; Scheurer et al., 2022; Ju et al., 2022; Bang et al., 2022; Liu et al., 2022) |
| Moral & normative judgements | (Forbes et al., 2020; Nahian et al., 2020; Jiang et al., 2022; Liu et al., 2022; Jin et al., 2022; Pyatkin et al., 2022; Liu et al., 2023c) |
| Others | *Sentiment and style transfer* (Ziegler et al., 2019; Peng et al., 2020; Liu et al., 2021a); *recipe generation* (Majumder et al., 2019); *predicting intent of emails* (Zhou et al., 2021); *machine translation* (Wang et al., 2021) |

Table 2: Articles categorised by target task.

ences or values can be contrasted with general investigations of model capabilities to estimate the so-called "alignment tax" (Liu et al., 2022). For instance, Korbak et al. (2023) rely on two metrics: (i) misalignment score, calculated using the same automated reward functions as training (toxicity score, number of PII instances per character, number of PEP errors per character), and (ii) capability score, calculated as the KL divergence of output distribution from a highly capable model (GPT-3). Some articles assess the drop in other performance measures on NLP benchmark tasks measuring truthfulness, toxicity or bias (Bai et al., 2022a; Ouyang et al., 2022).

## D Articles with Other Contribution Types

In the main paper, we discuss papers that seek to embed, train or align LLMs with human preferences and values. Here, we give a brief overview of the other categories of papers which are excluded from the main review.

**Predict** These articles include detecting moral content from tweets (Hoover et al., 2020; Asprino et al., 2022) or adapting to moral shifts (Huang et al., 2022); predicting values and ethics from social media content (Maheshwari et al., 2017) or music preferences (Preniqi et al., 2022); linking event

or entity extraction with moral values in knowledge bases (Lin et al., 2017; Li et al., 2019); and identifying human values in arguments (Kiesel et al., 2022).

**Evaluate** These articles include those that benchmark judgements in moral or ethical situations (Tay et al., 2020; Hendrycks et al., 2020; Lourie et al., 2021; Ziems et al., 2022; Mirzakhmedova et al., 2023); assess social biases or social reasoning (Sap et al., 2019, 2020); evaluate performance on personality-aware dialogue (Zhang et al., 2018) or empathetic dialogue (Rashkin et al., 2019); and detect non-human identity in conversations (Gros et al., 2021). Others directly evaluate the values or traits of existing models (Schramowski et al., 2019; Jentzsch et al., 2019; Perez et al., 2022).

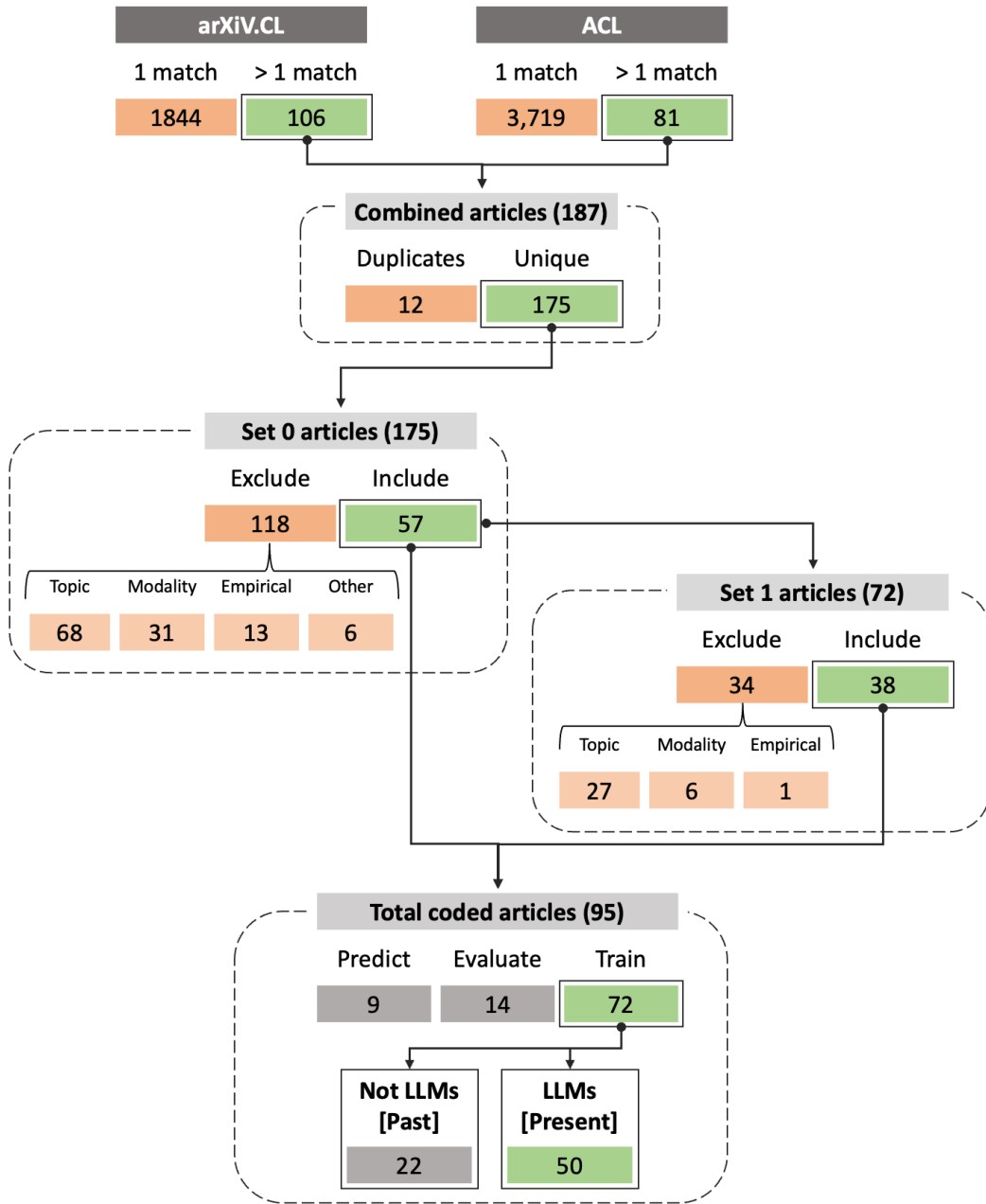

Figure 1: **Flowchart of the selection process for articles in our review.** We first match on the keywords defined in Tab. 1, keeping only articles with >1 matches in the title. We then de-duplicate articles posted on both arXiv and ACL. This initial set is called $S_0$. We apply the inclusion criteria listed in §2.1, and also add any possibly relevant references to the snowballed set ($S_1$) regardless of publishing venue. We also apply the inclusion criteria to $S_1$. Finally, we make two additional categorisations — the dominant contribution type of the article (predict, evaluate or train) and whether it uses LLMs.

| METADATA | | |
|---|---|---|
| **relevance** | Whether to include or exclude article from set | single choice: [exclude, include] |
| **exclusion criteria** | Reason for excluding the article | single choice: [topic, modality, empirical, other] |
| **exclusion detail** | Additional text summary of exclusion criteria reasoning | free-text |
| **snowball keys** | The bib keys of additional references | free-text |
| **contribution type** | Dominant contribution type of article | single choice: [evaluate, predict, train, other] |
| **contribution detail** | How are the main contributions of the article described? | free-text |
| **about LLMs** | Whether the article uses LLMs | single choice: [yes, no] |
| **short summary** | 1-3 sentence summary of the article | free-text |
| CONCEPTUAL THEME | | |
| **terminology** | Is feedback discussed using the terms 'preferences' or 'values'? | single choice: [ preferences, values, mix, other] |
| **motivation** | What is the motivation for feedback learning? | free-text |
| **target concepts** | Which human values or preferences are prioritised or included? | free-text |
| **concept defs** | How are human values or preferences defined? | free-text |
| **theories** | What theories (if any) are used to define preferences/values? | free-text |
| **concept scope** | Are concepts defined as universal or culturally/contextually understood? | free-text |
| **interpretation freedom** | What level of freedom are humans given in interpreting the in-scope target concepts, e.g., "helpfulness"? | single choice: [prescriptive paradigm, subjective paradigm, unclear] |
| LABOURAL THEME | | |
| **feedback generation** | How is feedback data collected? | multi choice: [human-generated explicit, human-generated implicit, model-generated, combined, other] |
| **feedback types** | What forms of feedback are collected? At what stage, and if for training or for evaluation? | free-text |
| **labour documentation** | Is the labour force documented? | single choice: [yes, no, nan] |
| **labour details** | What level of documentation or what details are documented? | free-text |
| **labour force** | Which human group(s) generate the feedback? | multi choice: [crowdworkers, in-house team authors, unknown] |
| **labour force detail** | What further detail is provided on who generates feedback? | free-text |
| **labour force size** | How many humans are involved in feedback collection for training and/or evaluation? | free-text |
| TECHNICAL THEME | | |
| **data size** | What is the size of the feedback dataset for training and for evaluation? | free-text |
| **intervention stage** | When and how is feedback integrated into the model? | multi choice: [pre-training, fine-tuning, prompting, other] |
| **metrics** | What metrics and which evaluation datasets are used? | free-text |
| **model approach** | Summarise the modelling methodology | free-text |
| PROCEDURAL THEME | | |
| **authorship** | Authorship composition of the article | single choice: [academia, industry, mixed] |
| **data availability** | Whether the data artifacts are publicly available | single choice: [yes, no, unclear] |
| **model availability** | Whether the model artifacts are publicly available | single choice: [yes, no, unclear] |
| **peer review** | Whether the article is peer-reviewed | single choice: [yes, no, unclear] |

Table 3: **Code book used for each article included in the review**. We show the field name, the prompt or instruction for the coder and the type of response variable (including options if it is a single or multiple choice question). Frequent communication between two coders was established to ensure the fields were being applied consistently.