# OpenReview forum: "The Past, Present and Better Future of Feedback Learning in Large Language Models for Subjective Human Preferences and Values"
_EMNLP/2023/Conference — EMNLP 2023 Main_

### Official Review · Reviewer_wQJD · 2023-08-01

**Soundness:** 4

**Excitement:**

4: Strong: This paper deepens the understanding of some phenomenon or lowers the barriers to an existing research direction.

**Missing References:**

In agreement with current methodology for literature selection.

**Paper Topic And Main Contributions:**

This paper conducts a review of 95 papers from either ACL or arXiv (cut-off in mid February of 2023) in which LLMs are aligned with subjective human preferences and values. The review is centered on a few axes, most importantly: before and after LLMs ("the past" and "the present"), as well as future perspectives; how feedback is collected and who it is collected from; and the different ways in which feedback has been utilized. The discussion revolves around five challenges identified across the selected literature, namely: non-universality of preferences and values; inconsistent conceptualization; feedback incompleteness; feedback operationalization; and representativeness of feedback sources.

**Questions For The Authors:**

1. In lines 651-655, authors argue that "researchers should consider techniques to model the noise and distribution of human feedback (...) rather than ignoring disagreement by aggregating responses"; soon thereafter, however, in lines 676-678, authors also warn that "[the lack of diversity within crowdworkers] results in a non-democratic feedback process"; my question is: if data sources remain non-representative of the population at large, wouldn't a finer-grained modeling of their feedback increase bias? In other words, finer-grained modeling is only a possibility *after* representativeness is properly addressed. For clarity, this dependency might be worth mentioning.

2. In lines 129-134, authors conceptualize LLMs as "any decoder-only or encoder-decoder model (...)" but then introduce BERT as "among the first LLMs." Considering that BERT is encoder-only, the conceptualization should be modified to accommodate that.

**Reasons To Accept:**

The review is well-written, clearly organized, and based on a thorough selection of papers. The topic is extremely relevant and fast-evolving, which motivates a high-quality snapshot of the current literature. In addition, authors are able to carry out a thoughtful discussion from the selected papers. Overall, this paper will likely help the community in analyzing the progress made in an important and timely topic.

**Reasons To Reject:**

I don't see good reasons for rejecting this paper. However, I do share questions and potential improvement notes in the fields below.

**Reproducibility:**

N/A: Doesn't apply, since the paper does not include empirical results.

**Reviewer Confidence:**

4: Quite sure. I tried to check the important points carefully. It's unlikely, though conceivable, that I missed something that should affect my ratings.

**Typos Grammar Style And Presentation Improvements:**

On line 472: "backpropagated" instead of "backprogated."

Throughout the paper, there are a few opportunities to normalize mentions to "language models" to "LLMs."

---

> ### Author Rebuttal · Authors · 2023-08-25
>
> We thank the reviewer for their detailed comments and areas for improvement. We are encouraged that they find the topic to be extremely relevant and fast-evolving, and see no good reasons for rejecting the paper! We will respond to questions and minor comments.
>
> ### Comment 1: Tension between granularity and representativeness of feedback
> This is an insightful comment, thank you. We agree that there is a tension between the granularity of feedback and the amount of noise or bias. In this case, it may be that more granular feedback exacerbates the issues with a non-representative sample. We will add a short discussion of this to the paper, particularly in line with your comment that finer-grained modeling may only be beneficial after representativeness is properly addressed.
>
> ### Comment 2: LLM definition
> Thank you for spotting this inconsistency in our definition. We will modify the definition to accommodate encoder-only models like BERT.
>
> ### Comment 3: Typos, Grammar, Style and Presentation Improvements
> Thanks for noticing the typo, we will fix that. We will also check for places where we can unify our terminology from language models to LLMs.

---

### Official Review · Reviewer_NxJU · 2023-08-06

**Soundness:** 4

**Excitement:**

4: Strong: This paper deepens the understanding of some phenomenon or lowers the barriers to an existing research direction.

**Paper Topic And Main Contributions:**

This survey and partially position paper offers a comprehensive overview of incorporating human feedback into LLMs, drawing from 95 papers. The paper focuses on the subjectivity of human feedback, encompassing aspects like definitions of human preference and values, annotation processes, and annotator distributions.

The primary contributions of this paper are twofold. First, through a thorough literature review, it provides a well-organized and insightful analysis of past and current research on how NLP leverages human feedback to enhance language technologies. Second, the paper highlights five crucial challenges and presents recommendations that can serve as valuable guidance for future research in this emerging field.


**Questions For The Authors:**

A. What do you mean by “Input” keywords and “output” keywords in Table 1? I could not find explanations in Section 2, nor in its caption.

**Reasons To Accept:**

- The paper presents a thorough and well-organized literature review.
- It effectively highlights the important challenges in incorporating human feedback into LLMs, raising awareness of the subjectivity and incompleteness of human preferences and values.


**Reasons To Reject:**

- Not much I can think of

**Reproducibility:**

4: Could mostly reproduce the results, but there may be some variation because of sample variance or minor variations in their interpretation of the protocol or method.

**Reviewer Confidence:**

4: Quite sure. I tried to check the important points carefully. It's unlikely, though conceivable, that I missed something that should affect my ratings.

**Typos Grammar Style And Presentation Improvements:**

- There are flipped double quotes in line 65.
- Some parts in the Conclusion (lines 734-740) were not discussed in the paper but were first introduced in the conclusion section. This feels a little awkward, as the title of the section does not include "Discussions."

---

> ### Author Rebuttal · Authors · 2023-08-25
>
> We thank the reviewer for their support of our contributions and are encouraged that they see no reasons to reject the paper! We will now respond to each of their questions and minor comments.
>
> ### Comment 1: Input and Output Keywords
> We agree that this was confusing terminology! We intended "input" column to correspond to the raw, unprocessed keywords (e.g. "alignment") and the "output" column to correspond to the stemmed, processed keyword (e.g. "align"). In some cases, keywords are unaltered (e.g. "value" is not stemmed to "valu") because stemmed versions return too many false positive matches. We will update the table terminology to describe this better and make sure any column headers are clearly explained in the caption.
>
> ### Comment 2: Typos, Grammar, Style and Presentation Improvements
> Thanks for these suggestions. We will correct the flipped double quotes. Regarding the Conclusions section, we will either (i) consider renaming it as "Discussions and Conclusions" to account for the wider scope; or (ii) find a way to rework the new comments on peer review and artefact availability earlier in the paper so they are not freshly introduced in the conclusion.

---

### Official Review · Reviewer_TDYC · 2023-08-10

**Soundness:** 4

**Excitement:**

4: Strong: This paper deepens the understanding of some phenomenon or lowers the barriers to an existing research direction.

**Paper Topic And Main Contributions:**

The paper is a survey/review focusing on the feedback mechanisms between humans and LMs. The analyzed papers are divided into the past (2014-2019) and the present (2019-today). The work primarily centers on the latter, initially examining how LLMs can better align with an optimum based on users' values and preferences. A relatively significant portion of the article delves into the conceptual difference between preference and value. Subsequently, it discusses current techniques for gathering direct and indirect feedback data, as well as the challenges associated with the lack of universality in values and preferences among users, resulting in noise in the inputs to LLMs. A final section addresses the issue of crowd workers generating feedback, often in limited numbers and carrying human biases that can prove to be unrepresentative and distorted relative to the population of end users.

**Reasons To Accept:**

* The process of paper selection is explained very clearly and transparently.
* The final pool is representative yet reasonably limited, indicating that each paper was given appropriate attention. The content of the survey supports this fashion as well.
* The survey demonstrates a good balance between methodological and technical discussion, comprehensively covering the topic concisely.
* The "Limitations" section is interesting on its own, adding depth to the work. The points expressed hold validity generally for surveys, extending beyond the context of this specific study.

**Reasons To Reject:**

The following are minor points and not actual reasons to reject.
* The sections "past," "present," and "future" are not very balanced. Specifically, the "past" section appears to be too condensed, while the "future" section is overly extensive considering the "survey" nature of the work.
* The paper often becomes more engaging in sections less tied to the survey (e.g., the "future" part). In the initial pages, some paragraphs contain too many citations - in extended format - that the actual text consists of brief, disjointed sentences (particularly on pages 4 and 5). Overall, there is tension between the survey nature and the aspiration for a broader scope.
* Each of the three sections would benefit from at least an initial contextualization paragraph before delving into the actual discussion (especially the "past" and "present" sections).

**Reproducibility:**

N/A: Doesn't apply, since the paper does not include empirical results.

**Reviewer Confidence:**

3: Pretty sure, but there's a chance I missed something. Although I have a good feel for this area in general, I did not carefully check the paper's details, e.g., the math, experimental design, or novelty.

---

> ### Author Rebuttal · Authors · 2023-08-25
>
> We thank the reviewer for their detailed comments and support of our contributions, with no stated reasons to reject the paper. We will now comment briefly on minor points for improvement, all of which we agree with and will make changes to the camera-ready version of the paper accordingly.
>
> ### Comment 1: Past vs Future section is unbalanced
> We agree that the "past" section is currently more condensed; it was originally reduced in length to meet page limits. With additional space for camera ready, we can bring back in some additional detail on papers falling into this section.
>
> The "future" section is longer because it also summarises key findings or methodological regularities from the review as whole, then uses these findings to make recommendations. As you suggest (see Comment 4), we will add an initial contextualisation paragraph to each section describing its purpose and scope.
>
> ### Comment 2: Paragraphs with dense citations
> We are unfortunately restricted in the citation format we can use by the EMNLP style guidelines. We agree that the extended format of citations as (Author, Year) and not numeric references ([1]) is clunky at times for a survey paper given the density of cites. In the camera-ready paper, we will devote some of the extra space where possible to expanding the actual text in paragraphs to reduce the clustering of citations.
>
> ### Comment 3: Tension between survey nature and broader scope
> As we mention in Comment 1, it is a dual aim of this paper to (1) survey the existing literature, (2) then draw on regularities, findings and critiques of this survey to also provide recommendations for future work. We will make sure the introduction appropriately communicates the intended scope, purpose and aims of the paper.
>
> ### Comment 4: Initial contextualisation paragraphs at the start of each section
> We thank the reviewer for this suggestion. We will certainly implement this in the camera-ready version as it will improve the paper and help to resolve Comment 1.

---

### Meta-Review · Area_Chair_BAQJ · 2023-09-10

**Recommendation:** 5

**Metareview:**

This paper is a survey on how human feedback is used for learning in large language models. The authors present a literature review as well as a set of challenges and recommendations for the future. The paper seems like a good fit for the theme track.

Pros:
* The literature review is thorough and the authors clearly describe their process for the review, including selection of papers, etc.
* The paper is well-organized and the challenges and recommendations mentioned can be useful to other researchers

Cons:
* The reviewers mention a few minor points related to clarity and wording, which can be improved for the camera-ready

---

### Decision · Program_Chairs · 2023-10-07

**Decision:**

Accept-Main

**Comment:**

This paper is a survey on how human feedback is used for learning in large language models. The authors present a literature review as well as a set of challenges and recommendations for the future. The paper seems like a good fit for the theme track.

Pros:
* The literature review is thorough and the authors clearly describe their process for the review, including selection of papers, etc.
* The paper is well-organized and the challenges and recommendations mentioned can be useful to other researchers

Cons:
* The reviewers mention a few minor points related to clarity and wording, which can be improved for the camera-ready